# Impact of a Moderately Hypocaloric Mediterranean Diet on the Gut Microbiota Composition of Italian Obese Patients

**DOI:** 10.3390/nu12092707

**Published:** 2020-09-04

**Authors:** Silvia Pisanu, Vanessa Palmas, Veronica Madau, Emanuela Casula, Andrea Deledda, Roberto Cusano, Paolo Uva, Sarah Vascellari, Francesco Boi, Andrea Loviselli, Aldo Manzin, Fernanda Velluzzi

**Affiliations:** 1Department of Biomedical Sciences, University of Cagliari, 09124 Cagliari, Italy; s.pisanu6@gmail.com (S.P.); vanessapalmas@hotmail.it (V.P.); veromadau91@gmail.com (V.M.); em.casula5@gmail.com (E.C.); svascellari@unica.it (S.V.); aldo.manzin@gmail.com (A.M.); 2Department of Medical Sciences and Public Health, University of Cagliari, 09124 Cagliari, Italy; andredele@tiscali.it (A.D.); boi@medicina.unica.it (F.B.); fernandavelluzzi@gmail.com (F.V.); 3CRS4, Science and Technology Park Polaris, Piscina Manna, Pula, 09010 Cagliari, Italy; robycuso@crs4.it (R.C.); paolo.uva@crs4.it (P.U.)

**Keywords:** gut microbiota, obesity, weight loss, Mediterranean diet, 16S rRNA, high-throughput sequencing

## Abstract

Although it is known that the gut microbiota (GM) can be modulated by diet, the efficacy of specific dietary interventions in determining its composition and diversity in obese patients remains to be ascertained. The present work aims to evaluate the impact of a moderately hypocaloric Mediterranean diet on the GM of obese and overweight patients (OB). The GM of 23 OB patients (F/M = 20/3) was compared before (T0) and after 3 months (T3) of nutritional intervention (NI). Fecal samples were analyzed by Illumina MiSeq sequencing of the 16S rRNA gene. At baseline, GM characterization confirmed typical obesity-associated dysbiosis. After 3 months of NI, patients presented a statistically significant reduction in body weight and fat mass, along with changes in the relative abundance of many microbial patterns. In fact, an increase in the abundance of several Bacteroidetes taxa (i.e., Sphingobacteriaceae, *Sphingobacterium*, *Bacteroides* spp., *Prevotella stercorea*) and a depletion of many Firmicutes taxa (i.e., Lachnospiraceae members, Ruminococcaceae and *Ruminococcus*, Veillonellaceae, *Catenibacterium*, *Megamonas)* were observed. In addition, the phylum Proteobacteria showed an increased abundance, while the genus *Sutterella*, within the same phylum, decreased after the intervention. Metabolic pathways, predicted by bioinformatic analyses, showed a decrease in membrane transport and cell motility after NI. The present study extends our knowledge of the GM profiles in OB, highlighting the potential benefit of moderate caloric restriction in counteracting the gut dysbiosis.

## 1. Introduction

The worldwide prevalence of obesity nearly tripled between 1975 and 2016. In 2016, more than 1.9 billion people aged 18 and over were overweight, and over 650 million of these were obese [1]. In the same year, more than a third of the Italian population was overweight (35.3%) and one person in ten was obese (9.8%) [2].

A BMI (weight in kilograms/height^2^ in meters) between 25 and 29.9 kg/m^2^ defines an overweight person, while obesity is determined by a BMI of >30 kg/m^2^. Increased BMI has been associated with an increased risk of all-cause mortality [3,4] and specific causes of mortality including cancer, cardiovascular, and respiratory deaths [4].

Obesity is a chronic disease characterized by a multifactorial etiology including genetic, behavioral, environmental, psychological, social, and cultural factors that result in a positive energy balance that promotes excessive fat deposition [5].

The gut microbiota (GM), the community of microorganisms colonizing the gastrointestinal tract, is needed for the proper development of infants and the maintenance of homeostasis and health throughout the life [6]. In fact, compositional changes of GM have been linked to metabolic disorders, including obesity and metabolic syndrome, and functional gastrointestinal disorders [7].

Several scientific pieces of evidence confirm that the metabolic activity of the intestinal microbiota can play an important role in the pathogenesis of obesity. Different mechanisms through which GM can promote the deposition of fat have been hypothesized, including the suppression of the fasting-induced adipose factor and the reduction in 5′-AMP activated protein kinase (AMPK) activation, with an increase in lipoprotein lipase activity, the extraction of energy from fiber, the changes in intestinal permeability and the consequent increase in Gram-negative bacterial lipopolysaccharide (LPS) and chronic low-grade inflammation, and the metabolism of bile acids [4,8,9].

Concerning the modifiable nature of the GM and the driven role of diet in determining its composition [10], the effects of many weight loss interventions on GM composition have been investigated in the last decade. A recent systematic review on this topic showed that both restrictive diets (very low energy or low-carb) and bariatric surgery (BS) interventions caused a decrease in the abundance of the microbiota, and generally reduced *Lactobacillus* spp. and *Bifidobacterium* spp., which are involved in the cross-feeding of butyrate-forming bacteria [11].

To date, elaborating effective strategies aimed at obtaining weight loss while simultaneously restoring the gut microbial composition represents an open challenge. In fact, if, on the one hand, beneficial modulation of GM can facilitate both the improvement of metabolic outcomes and weight-control in the long term, on the other, negative impacts on the already compromised gut microbial balance can have a deleterious effect on the colon. This consideration could identify a potential cause of the higher relapse rates registered after drastic diets [12].

In the present study, we investigated the potential role of a nutritional intervention (NI) based on moderate caloric restriction (duration = 3 months) in modulating GM composition and diversity in a group of overweight and obese patients (OB).

## 2. Materials and Methods

### 2.1. Patients Recruitment and Samples Collection

This study was approved by institutional review boards and human subject committees at the participating institutions (Prot.PG/2020/2973). All subjects gave written informed consent in accordance with the Declaration of Helsinki.

We recruited 23 OB patients, consecutively enrolled among obese outpatients at the Obesity Center of the University Hospital of Cagliari (Sardinia, Italy). Subjects of both sexes aged at least 18 were included. The inclusion criteria for the OB group were a BMI ≥25 and being “diet-free”. We defined “diet-free” as patients who did not follow any specific diet within the 12 months prior to recruitment in order to define the GM during their usual dietary habits. Exclusion criteria were the following: therapy with antibiotics, proton pump inhibitors, or metformin in the last 3 months; the use of prebiotics, probiotics, or dietary supplements in the last 3 months; the presence of intestinal bowel disease (IBD); history of cancer; diagnosis of psychiatric disorders. A group of 46 healthy normal-weight Sardinian subjects (NW), matched for gender and age, was recruited as the control at baseline. As well as OB patients, NW subjects were “diet-free” and did not report bodyweight changes in the last 2 years.

Stool samples from each subject were collected at outpatient facilities and delivered to the laboratory within 3 h. Fresh samples were stored at −80 °C until further processing. Samples from OB patients were collected both at baseline, before beginning the NI (T0), and after three months of NI (T3); samples from NW subjects were collected only at baseline.

### 2.2. Anthropometric and Nutritional Assessment

All the anthropometric measurements were collected on the same day as sample collection. Height, expressed in centimeters, was measured with a stadiometer. Body weight, expressed in kilograms, was measured with an impedance scale (Tanita BC-420, Tanita, Amsterdam, The Netherlands), which was also used to analyze body composition. The impedance analysis was executed at room temperature, with patients in a fasted state (for at least 2 h) and having had performed moderately intense physical activity at least 24 h prior. The following parameters of body composition were extracted and collected for the present study: bodyweight expressed in kilograms, BMI, fat mass expressed in kilograms and percentage, muscle mass expressed in kilograms, and basal metabolic rate (BMR). BMI was calculated by the ratio of weight in kilograms and height in meters squared. Waist circumference, expressed in centimeters, was measured following guidelines issued at the Airlie conference [13].

A weighted 3-day food intake record (3d-FR, including 2 days of the week and 1 weekend day) was registered on the same day of sample collection. The analysis was performed with the Winfood^®^ software (v. 3.0) and the following parameters were obtained: average daily caloric intake expressed in Kcal, daily percentage of macronutrients intake (carbohydrates, lipids), daily intake of proteins in grams, daily percentage of saturated lipids intake (of the total lipids intake), and daily intake of fiber in grams. Overall dietary habits were evaluated through the Mediterranean Diet Score (MedDietScore, range 0–55), a composite index of eleven food items that assesses adherence to the Mediterranean diet (MD) [14], where higher scores indicate higher compliance. One of the eleven items evaluates alcohol consumption, assigning a maximum score of 5 for a consumption <300 mL/day, corresponding to the intake suggested by the Mediterranean model, and a minimum score of 0 for both a consumption >700 mL/day or none [14]. According to the MedDietScore, we defined “moderate” as a consumption of alcohol lower than 300 mL per day and “none” as the absence of alcohol consumption; we also defined “rare” as the occasional consumption of alcohol (maximum 1 time per week).

### 2.3. Nutritional Intervention

Patients were instructed to follow a prescribed dietary plan, with a daily caloric intake equivalent to their BMR (±10%), as detected by the impedance analysis. The diet consisted of a 7-day meal plan (three meals and two snacks), with indications of the various food and their weighted intake expressed in grams, and a balanced composition of macronutrients (carbohydrates 55%, lipids 25%, protein 20%; fiber ≥25 g/day), as recommended by the Reference intake levels of nutrients and energy for the Italian population (LARN) guidelines [15]. Vegetables, fruit, cereals, fish, and pulses, typical of the Mediterranean style, were included in the diet. In particular, the following nutritional intakes were included in the diet: 5 portions per day of fruits and vegetables; 3 portions per week of poultry; limited intake of red meat (maximum 3–4 times per month); 2 portions per week of eggs; 2 portions per week of dairy products; 4 portions per week of fish; 3 portions per week of pulses. Patients were advised to use extra-virgin olive oil as seasoning, to increase the consumption of whole grains, and to avoid added sugar and industrial foods [14].

### 2.4. Microbial DNA Extraction and 16S rRNA Gene-Based Illumina MiSeq Sequencing

DNA extraction, purification, and quantification by real-time PCR were performed as previously described [16]. Barcoded amplicon libraries for the bacterial community analysis were generated using primers targeting the V3-V4 hypervariable region of the bacterial 16S rRNA gene and the Nextera XT index kit (Illumina, Inc., San Diego, CA, USA).

### 2.5. Bioinformatics and Statistics

Analysis of the data generated on the MiSeq System was carried out using the BaseSpace 16S Metagenomics App (Illumina), whereas mapping of the operational taxonomic units (OTUs) to the Greengenes database (v.13.8) was performed using the Quantitative Insights Into Microbial Ecology (QIIME) platform (v.1.8.0), clustered into 97% identity using a two-step picking open-reference operational taxonomic unit (OTU).

Alpha diversity was generated with the script alpha_rarefaction.py in QIIME to obtain the Shannon index. Alpha diversities were compared by using the Wilcoxon test for paired data. Beta diversity was generated in R-vegan, using the Bray–Curtis distance. The statistical significance of beta diversity was determined with Permutational Multivariate Analysis of Variance (PERMANOVA) (R-vegan, function adonis). A *p*-value (*p*) ≤ 0.05 was considered statistically significant; a *p* between 0.05 and 0.1 was considered a tendency.

The analysis of the taxonomic profiles was performed in R software v.3.5.2. At baseline, a comparison between OB and NW was performed considering only bacteria present in at least 25% of our samples and with a relative abundance ≥0.1% in cases and/or controls, by using the Kruskal–Wallis test (KW) followed by false discovery rate (FDR) adjustment. Linear Discriminant Analysis Effect Size (LEfSE) was employed for the identification of biomarkers, including only bacteria identified as significant in the KW test, after FDR correction. The comparison between gut microbiota profiles before and after the intervention was performed by using the Wilcoxon test for paired data. Only bacteria present in at least 25% of our samples and with a relative abundance ≥0.1% before or after intervention were considered. All the *p*-values were adjusted for FDR. Q-values (q) < 0.05 were considered as statistically significant. Anthropometric measurements and nutritional data in OB patients before and after NI were compared using a *t*-test for paired data. Differences in the anthropometric data, alpha diversity, and Firmicutes/Bacteroidetes ratio before and after NI by BMI categories were evaluated by one-way ANOVA with repeated measures. The comparison of the ratio Firmicutes/Bacteroidetes in different BMI categories of OB patients at baseline was evaluated by one-way ANOVA for independent samples. A post hoc analysis was conducted only if the omnibus test was statistically significant (*p* < 0.05).

The comparison of nutritional data between OB and NW was performed using a *t*-test for independent samples.

Phylogenetic Investigation of Communities by Reconstruction of Unobserved States (PICRUSt) [17] (v. 1.1.4) was performed on the Galaxy computational tool to infer metagenome composition in the samples. QIIME pipeline was used for OTUs picking from data generated on the Illumina platform. After the OTUs’ normalization by copy number, metabolic pathways were predicted and classified by the Kyoto Encyclopedia of Genes and Genomes (KEGG) pathway database orthologs, at hierarchical level 3. OTUs’ contribution to the abundance of functional categories in each sample was detected by the function “metagenome contribution by higher category”, using KEGG’s classification type. The statistical significance of differences in metabolic pathways before and after NI was analyzed for all metabolism pathways that presented mean differences (the mean after NI minus the mean before NI) of at least 0.1% after the intervention, using the Wilcoxon test for paired data (R software v. 3.5.2) with Benjamini and Hochberg correction (FDR cutoff < 0.05).

### 2.6. Availability of Data and Materials

Our sequence data for the 16S rRNA gene were deposited in the European Nucleotide Archive (ENA) (https://www.ebi.ac.uk/ena), under the study accession number PRJEB39062 (http://www.ebi.ac.uk/ena/data/view/PRJEB39062).

## 3. Results

Among the OB patients, three (13%) showed a BMI between 25 and 29.9 kg/m^2^ (overweight), eight (35%) showed a BMI between 30 and 34.9 kg/m^2^ (class I obesity), nine (39) showed a BMI between 35 and 39.9 (class II obesity) kg/m^2^, and three (13%) showed a BMI ≥ 40 kg/m^2^ (class III obesity). Most of the included patients had metabolic dysfunctions: seven patients (30%) had hypertension, six (26%) patients had dyslipidemia, and five (22%) patients were diagnosed with diabetes or insulin resistance. Six patients (26%) were active smokers and six patients (26%) reported a moderate alcohol consumption. NW subjects did not present any of the metabolic dysfunction observed in OB patients; 7 (15%) were active smokers and 12 (26%) reported a moderate alcohol consumption. Table 1 summarizes the clinical characteristics of OB patients and NW subjects.

The nutritional anamnesis in OB before NI highlighted an excessive consumption of processed meat, industrial food, and sugary drinks, and a low consumption of fruit, vegetables, and pulses, as well as the absence of whole grains (Table 2). The MDS at baseline was 29 ± 5, significantly lower than that in NW (33 ± 4; *p* = 0.001) indicating a low adherence to the Mediterranean Diet. On the other hand, NW subjects followed a healthier diet, in terms of both caloric intake, composition of macronutrients, and fiber intake, which were more in line with LARN guidelines [15].

At the time of the second sample collection, at T3, the 23 OB patients showed a statistically significant decrease in body weight (−7.5%), BMI (−4.5%), waist circumference (−5.6%), and fat mass (−13.5%), without variations in muscle mass. We additionally investigated the differences in anthropometric data before and after the intervention, dividing the sample by BMI categories, as shown in the Appendix A, confirming the decrease in body weight, waist circumference, and fat mass for each subgroup. In particular, patients with class I obesity showed a greater improvement in body composition. In addition, the analysis of the 3-d FR, reporting the nutritional intake in the 3-days before the second sample collection, showed a decrease in the caloric intake compared with baseline, and a lower, although not significant, intake of saturated lipids, along with a higher intake of fiber, without variation in carbohydrates intake, indicating a good adherence to the diet. Moreover, at T3, OB patients proved a greater adherence to the Mediterranean diet, though not significant, presenting a similar mean value than NW (T3 = 32 ± 5; NW = 33 ± 4, *p* = 0.162).

The anthropometric measurements and the nutritional intake before and after intervention are shown in Table 2.

### Gut Microbiota Diversity and Composition

#### 3.1.1. Gut Microbiota Diversity

No significant difference in the Shannon index was observed between OB patients at T0 and controls (OB = 3.41 ± 0.54, NW = 3.46 ± 0.36, *p* = 0.834). Similarly, no significant change in this index (3.39 ± 0.66, *p* = 0.761) was found when comparing T0 and T3 samples of OB patients. Differences in the Shannon index across different BMI categories were not statistically significant, both at T0 and T3 (p = 0.092 and 0.991, respectively). However, a significant increase in the Shannon index was found in patients with class I obesity (T0 = 3.49 ± 0.33, T3 = 3.52 ± 0.33, *p* = 0.007) (Appendix A).

On the other hand, the PERMANOVA analysis of beta diversity showed significant separation between OB and NW at T0 (*p* = 0.002, F = 4.916, R^2^ = 0.068), as illustrated in Figure 1a.

Notably, the GM of OB did not segregate from that of NW after the NI (Figure 1a, *p* = 0.122). The matched paired analysis in OB, before and after NI, showed a trend toward separation (*p* = 0.100), though statistical significance was not reached.

As for alpha diversity, when restricting the analysis to OB patients, no statistically significant differences were noticed across different BMI categories, both at T0 and T3 (*p* = 0.539 and 0.649, respectively), suggesting that the variability in microbial diversity among the included patients was not driven by individual differences in weight (Appendix A). Similarly, no significant differences were found between females and males both at T0 and T3 (Appendix A).

#### 3.1.2. Gut Microbiota Composition

The enrolled OB patients had a Firmicutes/Bacteroidetes ratio more than twice that of controls (OB = 4.73 ± 5.26, NW = 1.75 ± 1.82, *p* = 0.007). After NI, a reduction in the ratio of Firmicutes/Bacteroidetes (M ± SD = 2.42 ± 2.86) was observed. However, the change was not significant (*p* = 0.128). Taking into account the wide range of BMI among the included patients, we compared the ratio among different BMI categories, both at T0 and T3, but no significant differences were observed (*p* = 0.092 and 0.991, respectively).

Based on the relative abundance of the gut bacteria, the LefSe algorithm was applied to the taxa significant in the KW test and confirmed after the FDR adjustment (Figure 1b): a total of 19 biomarkers were associated with obesity and overweight, with the phylum Firmicutes at the top of the rank. Three families belonging to Firmicutes were also enriched in OB (Gemellaceae, Streptococcaceae, and Thermicanaceae; q = 0.0026, 0.0039, and 0.0305, respectively), along with the genera *Megamonas*, *Gemella*, *Streptococcus*, *Megasphaera*, *Veillonella*, *Thermicanus*, and *Catenibacterium* (q ranging from 0.0006 to 0.0412), and the species *Megamonas funiformis*, *Megasphaera hominis*, and *Eubacterium biforme* (q = 0.0305, 0.0005, and 0.0286, respectively). In addition, two genera belonging to Gammaproteobacteria, known for pro-inflammatory activity (*Enterobacter*, *Serratia*), were more abundant in OB (q = 0.0092 and 0.0112, respectively). On the contrary, 19 biomarkers were associated with leanness, with Bacteroidetes at the top, followed by many corresponding taxa: we found an increase in the families Flavobacteriaceae, Porphyromonadaceae, and Sphingobacteriaceae (q = 0.0173, 0.0095, and 0.0005, respectively), the genera *Rikenella*, *Flavobacterium*, *Parabacteroides*, and *Pedobacter* (q ranging from 9.9 × 10^−4^ to 0.0393), and several species belonging to these genera. We repeated the same analysis at T3: the differences with NW controls were fewer in terms of both number (38 significant results at T0, 22 significant results at T3) and strength. As shown in Figure 1c, differences in the Firmicutes/Bacteroidetes ratio and in the relative abundance of same taxa within these phyla were no longer significant, as well as some taxa within them. In addition, *Desulfovibrio piger*, belonging to Proteobacteria, was associated with OB at baseline, but not after NI.

The relative abundance of several microbial patterns was significantly modified after NI (Figure 2). When considering the FDR adjustment, we obtained a total of 24 significant results. The median values and interquartile range (IQR) and the corresponding *p*-values and q-values for each significant result are shown in Table 3. The comparison between OB (at T0 and T3) and NW, as well as the matched analysis, were performed considering only female patients. The results are shown in the Appendix A and partially confirmed the trends observed in all OB samples, within the phyla Firmicutes and Bacteroidetes. Additionally, in females, we also found an increase in Actinobacteria and its members. In line with the findings obtained in all OB patients, many biomarkers were no longer significant at T3.

Following NI, an increase in Proteobacteria (q = 0.001) was observed, while the *Sutterella* genus (q = 0.002), belonging to the same phylum, decreased significantly. One family within the Bacteroidetes phylum was enriched (Sphingobacteriaceae, q = 0.003), along with its species *Sphingobacterium shayense* (q = 0.003). An increase in *Bacteroides uniformis* and *Prevotella stercorea* (q = 0.036 and 0.001, respectively), within the same phylum, was also observed. On the other hand, two families belonging to the Firmicutes phylum were depleted (Ruminococcaceae and Veillonellaceae, q = 0.001 and 0.004), as well as *Ruminoccocus* and *Ruminoccus* spp. (q ranging from 0.001 to 0.012). In addition, the abundance of several Firmicutes belonging to the Lachnospiraceae family changed significantly, with an increase in *Coprococcus eutactus* (q = 0.005), as well as a decrease in *Roseburia*, *Roseburia faecis*, and *Pseudobutyrivibrio xylanivorans* (q ranging from 0.001 to 0.026). Moreover, the relative abundance of *Megamonas* and *Megamonas funiformis*, belonging to Firmicutes, decreased after intervention (q = 0.046 and 0.038, respectively). However, an increase in the Firmicutes genera *Catenibacterium* and *Veillonella* (q = 0.049 and 0.004, respectively) was also observed, together with the increase in *Sedimentibacter hydroxybenzoicus* and *Veillonella montpellierensis* (q = 0.004 and 0.001, respectively).

#### 3.1.3. Predicted Metabolic Pathways

After NI, a significant difference in seven metabolic pathways was found: one pathway related to membrane transport (ABC transporters), one pathway associated with transporters, two pathways related to cell motility (“flagellar assembly”, “bacterial motility proteins”), and one pathway associated with sporulation decreased. On the other hand, the pathways “lipopolysaccharides biosynthesis proteins” and “membrane and intracellular structural molecules” increased after intervention (Figure 3).

## 4. Discussion

The present work compared the GM of 23 overweight and obese patients before and after 3 months of NI, aimed at inducing weight loss. At the time of the second sample collection, the patients presented a significant decrease in body weight, waist circumference, and fat mass. The patients followed a moderately low-calorie diet, which allowed muscle mass to be preserved. In fact, an efficient dietary approach should focus not only on changes in the bodyweight but also on body composition, considering that the reduction in bodyweight can be driven not only by a loss of fat mass but also by alterations in both the fat-free mass and fluid. Moreover, the decrease in muscle mass can be connected with lowered resting energy expenditure/metabolism, fatigue, declines in neuromuscular function, and increased risk of injuries [18]. Furthermore, the diet was based on the Mediterranean regimen in free-living conditions. The Mediterranean diet is known not only to be efficient for weight loss and cardiovascular risk level reduction in OB individuals, but also to positively modulate GM composition and diversity [19,20,21]. Moreover, it has been proposed that MD represents an important field of investigation as regards its bacterial-targeted action and a better understanding of its effects would be useful not only for the development of targeted bacterial therapies, but also for the prevention of food-related diseases [22].

At baseline, OB showed a low adherence to MD (MedDietScore = 29 ± 5), significantly different from that of NW, which, instead, showed good adherence (MedDietScore = 33 ± 6) [14]. The same mean score observed in NW in the present study was found in a large multinational study, which recruited participants from 20 Mediterranean islands, including Sardinia [23]. As detected by the impedance analysis, OB showed a significant decrease in body weight, waist circumference, BMI, and fat mass after NI as well as greater adherence to MD (MedDietScore = 32 ± 5), which, although not changing significantly (*p* = 0.665), became closer to that observed in NW (*p* = 0.162). It should be noted that, as reported in the literature, the use of MedDietScore for the assessment of adherence to MD may also have some limitations due to the fact that it does not take into account the weight of food items (except for alcohol) [14], although in our study, food diaries were also collected. In the present work, the analysis of both MD questionnaires and food diaries showed that patients, despite having largely complied to the prescribed reduction in caloric intake, did not completely change their eating habits in terms of foods frequency. More specifically, the consumption of some food typical of MD was less frequent than recommended (fish, pulses, whole grains), although higher than the one detected at baseline. It can be hypothesized that a longer follow-up period could have allowed us to obtain a more complete change in eating habits as well as greater adherence to MD.

MedDietScore was significantly correlated with the abundance of several bacterial taxa in previous studies [24,25,26]. Coherently, the abundance of several microbial profiles was significantly different in OB compared to NW at baseline, and this difference can be summarized by the higher ratio of Firmicutes/Bacteroidetes in OB, with the trend being confirmed also at the lower taxonomic levels, and by the high content of Gammaproteobacteria. The higher ratio of Firmicutes/Bacteroidetes and the identified microbial taxa within these two phyla can be associated with an increased hydrolysis of non-digestible polysaccharides, and with increased nutrients absorption [27], resulting in an increase in calorie production. In addition, the higher relative abundance of Firmicutes has been found to raise the number of lipid droplets and the export of fatty acids to the liver in the animal model [27,28]. Regarding the identified alterations within the phylum Bacteroidetes, and in particular, the depletion in Flavobacteriaceae, Porphyromonadaceae, and Sphingobacteriaceae, findings from the animal model have demonstrated the capacity of lifestyle interventions, based on specific diets or increased physical activity, to restore the normal contents in the gut [29,30]. However, the mechanisms explaining the Bacteroidetes modulation of the bodyweight remain still unclear, even though a recent experiment on mice has suggested that the anti-obesity properties may be connected with the production of succinate and secondary bile acids, which activate the intestinal farnesoid X receptor (FXR) signaling pathway, with a consequent stimulation of intestinal gluconeogenesis, along with the increase in the gut barrier integrity [31]. Lastly, lipopolysaccharides from Gram-negative bacteria, including enterobacteria, cause an increase in intestinal permeability, leading to low-grade inflammation, called endotoxemia, frequently observed in obese subjects [32]. When taking into consideration only female participants (20 OB patients compared with 40 NW controls), we confirmed the trends related to Firmicutes and Bacteroidetes phyla. At the same time, we identified additional biomarkers, including Synergistetes phylum, depleted in OB females, and Actinobacteria members, enriched in OB females. Synergistetes members are frequently found in anaerobic ecosystems, although in low abundance. In vitro models have demonstrated that these bacteria are acetate-utilizing and amino-acids degraders, but the association with obesity remains to be ascertained [33]. The higher abundance of Actinobacteria in OB patients compared with NW is in line with a previous study [34], even if other studies on OB population described a depletion in its genus *Bifidobacterium* [8], which had shown anti-obesity proprieties at the species level [35].

No significant difference in alpha diversity was observed between OB and NW at baseline, and before and after NI in the OB group. This findings are partly in contrast with those of previous studies; in fact, two meta-analyses found a decrease in the gut richness in obese patients [36,37], while two other meta-analyses did not observe a significant association between obesity and alpha diversity [38,39]. Taking into account the contrasting results in the literature, Stanislawski et al. recruited healthy subjects of different ethnicities for evaluating the impact of race/ethnicity in modulating the association between GM and obesity: surprisingly, alpha diversity was significantly decreased among Black subjects, but not in White subjects [40], thus suggesting that the association between obesity and alpha diversity may be restricted to specific ethnicities, and that further studies focused on how population heterogeneity influences the relationship between the GM and obesity are required. Despite NI, the alpha diversity of the patients remained constant over the study period, in line with other 15 intervention studies included in a recently published review: the nutritional interventions, lasting from 5 days to 24 weeks, focused on the increased intake of high-fiber food in modulating the GM of healthy participants. In line with our work, the interventions had no effect on alpha diversity, although bacterial relative abundance changed significantly [41]. However, in the present study, a significant increase in the Shannon index was observed in patients with class I obesity (*p* = 0.007), who reported a more marked improvement in body composition and in particular a more marked reduction in waist circumference compared with the other BMI groups (*p* = 0.054).

The beta diversity analysis showed significant differences in microbial abundance between OB and NW at baseline (*p* = 0.002). Interestingly, the beta diversity value no longer varied significantly between the same NW controls and OB patients after NI (*p* = 0.122). In line with these findings, the paired analysis of beta diversity in OB patients before and after NI indicated a trend toward significance. Taken together, the results on microbial diversity suggest that a period of three months may not be sufficient for achieving a substantial and statistically significant change in the diversity of the gut community, although the comparison with NW suggested that several rearrangements in microbial community structure took place over the study period and although we observed significant increase in alpha diversity in patients with the greatest anthropometric improvement after NI.

At baseline, LDA showed a higher abundance of 19 taxa in OB compared to NW, while that of the other 19 taxa were lower in the same subjects. Noteworthy, only 13 out of 19 higher taxa were still significantly altered in OB after NI compared to NW; on the other hand, only 9 out of 19 lower taxa diverged significantly between OB and NW at T3. These data indicate that NI had a beneficial impact on intestinal microbial homeostasis limited to some bacterial taxa. This result could be explained, in part, by the not significant increase in MD adherence of OB after NI and, to a greater extent, by the evidence that alterations in eating habits represent an important modulating factor, although not the only one, of intestinal microbial composition, which works in conjunction to numerous factors both external and internal to the host [42].

In this study, NI showed a positive effect on the modulation of bacterial taxa, especially in those belonging to the phyla Firmicutes and Bacteroidetes. In detail, paired analysis showed a total of 24 significant results (1 at the phylum level, 3 at the family level, 7 at the genus level, and 13 at the species level) after FDR adjustment.

Regarding the gut microbial alterations within the phylum Bacteroidetes after NI, we observed an increased abundance in the family Sphingobacteriaceae and its species *S. shayense*, and a rise in *B. uniformis*. All of these three taxa were identified as being negatively associated with obesity at baseline. Remarkably, the administration of *B. uniformis* strains improved metabolic and immune dysfunction associated with intestinal dysbiosis in obese mice [43]. We also observed a depletion of Firmicutes taxa known for being associated with obesity (i.e., Veillonellaceae, *Megamonas*, and *M. funiformis)* that express propionate production pathways [44,45].

Walker et al. investigated the impact of three different diets on modulating the GM composition of 14 obese men. Volunteers were provided successively with a control diet, diets high in resistant starch (RS) or non-starch polysaccharides (NSPs), and a reduced carbohydrate weight loss (WL) diet, over the course of ten weeks [39]. The findings of Walker et al. suggested that supplementation with RS can balance the reduction in the Lachnospiraceae family and/or its members, caused by the WL diet. Interestingly, this reduction was also found after different types of hypocaloric diets (high protein, fiber-rich, or with prebiotics supplementation) in other studies [46,47,48].

Similarly, we found a decrease in Lachnospiraceae members (*Roseburia*, *R. faecis*, *P. xylanivorans*), although an increase in *C. eutactus*, within the same family, was also observed in the present work. Members of this family can hydrolyze starch and other sugars to produce butyrate and other short-chain fatty acids (SCFAs) and play a central role in the mechanisms of bacterial cross-feeding [49]. In particular, the genus *Roseburia* is involved in the control of gut inflammatory processes, atherosclerosis, and maturation of the immune system [50]. However, it should be noted that despite the well-known benefits provided by the members of this family, Lachnospiraceae was positively associated with metabolic diseases in humans and animal models [51,52,53,54]. At the same time, higher SCFAs production (acetate, propionate, and butyrate) was associated with gut dysbiosis and obesity in a recent study with a large sample size [55], and the finding was confirmed in a random effect meta-analysis [56].

In the present study, the changes in GM composition after the NI suggest a decrease in SCFAs producing bacteria (Lachnospiraceae and Veillonellaceae, *Ruminoccocus* spp., and *Megamonas*). It is unclear whether the beneficial effect of SCFAs is somehow compromised in obese subjects, or the effect is simply not strong enough to compensate for an incorrect lifestyle and/or genetic predisposition [57]. SCFAs are known to activate G protein-coupled receptors (GPCRs), including GPR41 and GPR43, expressed in human adipocytes, colon epithelial cells, and peripheral blood mononuclear cells [58]. It has been suggested that in the case of the obese condition, the binding of SCFAs to G protein-coupled receptors at the intestinal level might be attenuated, leading to increased intestinal energy harvesting and hepatic lipogenesis [56].

Noteworthy, we found an increase in *O. eae*, within *Oscillospira*, after the NI, although this was not confirmed after FDR adjustment. The genus *Oscillospira* has been associated with low BMI/leanness in several studies [51,59]. These bacteria metabolize glucoronate, a sugar found on the cell surface and in the extracellular matrix of most human tissues [60]. The degradation of host glucoronate by *Oscillospira* causes an energy expenditure for the host that may explain its association with leanness [59].

Cancello et al. evaluated the efficacy of a short-term dietary intervention on the GM of elderly Italian women with obesity: after 15 days of hospitalization following a hypocaloric Mediterranean diet, they noticed a decrease in pro-inflammatory bacteria, along with a moderate weight loss and improved metabolic function. In line with our study, the diet provided a daily energy deficit equal to 250 kcal. The study showed the efficacy of a balanced diet with moderate caloric restriction, even of short duration, in improving gut health, reversing the GM dysbiosis found at baseline [61].

When comparing our findings with those of weight loss strategies different than diet, bariatric surgery (BS) should be mentioned. This intervention is based on surgery on the stomach and/or intestine, aimed at losing weight. BS is an option for obese patients with a BMI above 40, or a BMI between 35–40 in the presence of comorbidities [53].

Several studies showed an increased relative abundance of the phylum Proteobacteria after BS [52,54,62,63,64]. Interestingly, this phylum was increased after NI in the current work. However, after BS, a rise in the class Gammaproteobacteria and in the pro-inflammatory genera *Escherichia*, *Klebsiella*, and *Enterobacter* has been reported [52,54,62,63,64,65], while they were not increased after the NI in the present work. Instead, a decrease in the *Sutterella* genus, which can degrade IgA antibodies and have been recently linked with gastrointestinal diseases [66], was found in our work. On the other hand, the reduction in the abundance of bacteria belonging to Proteobacteria phylum has been associated with higher adherence to MD [19,26]. In our study, these data were not confirmed, despite greater adherence of OB to MD over time. It must be considered that the mentioned studies analyzed a cohort of apparently healthy subjects, suggesting that the effect of MD on Proteobacteria and the related taxa could be different or ineffective if the obese population is evaluated. The increase in these taxa in OB following NI could be the indirect effect of diet on the reciprocal interactions of microorganisms resulting from intestinal microbial rearrangement, although the physiological significance of this event must be further investigated.

However, an increase in the relative abundance of the phylum Verrucomicrobia or its members, an increase in members of the phylum Bacteroidetes assigned to the genus *Alistipes*, as well as a general decrease in members of the phylum Firmicutes were also reported after BS [54,62,63,64,65]. The decrease in Firmicutes members is in common with the one reported in this work. In addition, Firmicutes taxa within the genera *Veillonella* were shown to be increased after BS [65] and after the NI in this study.

Regarding the predicted metabolic pathways, a decrease in gene functions related to membrane transport and cell motility was shown after the NI in the present work. Increased cell motility has been recently associated with obesity [67] and metabolic syndrome [68], and observed in high-fat fed mice [69]. However, a rise in bacterial cell motility was registered in obese patients after BS [63]. We also observed an increase in “lipopolysaccharides biosynthesis proteins” and “membrane and intracellular structural molecules” after the NI. The lipopolysaccharides biosynthesis proteins were identified as associated with obesity in other recent studies [70,71], but not confirmed in a large recent study [72], and the direction of the association remains unclear.

The present study highlights the potential benefit of a moderately restrictive nutritional approach based on the Mediterranean model in counteracting the gut dysbiosis, commonly observed in obese and overweight patients. A key strength of the present study is that all the participants belonged to a restricted geographical area (Cagliari, Sardinia, Italy), with consequent reduction in interindividual variability. A major limitation of our work is that the microbiota characterization of the NW subjects was performed only at baseline. For this reason, the comparison between OB patients and NW controls was repeated after the intervention, using the NW data collected at baseline. This practice relies on the assumption that controls, whose body weight was stable for the past two years, did not change their lifestyle during the study period. However, it is possible that modifications in lifestyle, even minor, could have occurred, affecting the composition and diversity of the GM during the three months of observation. Another limitation is represented by the short follow-up period. It remains to be ascertained whether the identified microbial changes are consistent over time and how their permanence can be influenced by lifestyle habits. Finally, we did not observe differences in the Firmicutes/Bacteroidetes ratio and in the GM beta diversity across different BMI categories, suggesting that the microbiota variability among the included patients was not driven by differences in BMI. However, the small number of patients in each group limited the statistical power of the analyses.

Regarding methodological issues, it should be pointed out that targeting 16S variable regions at the species level cannot achieve the taxonomic resolution reached by sequencing the entire gene (around 1500 bp). This limitation should be taken into account when interpreting the results at the species level.

## Figures and Tables

**Figure 1 nutrients-12-02707-f001:**
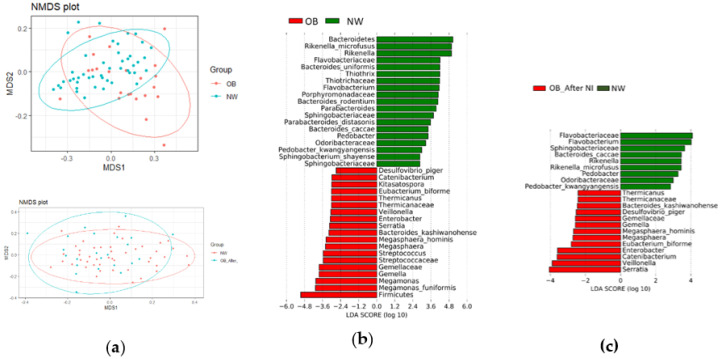
Diversity between obese (OB) and normal weight (NW) patients (**a**) Overweight and obesity are associated with altered beta diversity at baseline but not after a hypocaloric Mediterranean diet followed for three months. Three-dimensional scatter plots, generated using the Non-Metric Multidimensional Scaling (NMDS) conducted on the community Bray–Curtis distance matrix, showed a significant separation between OB patients and normal-weight controls at baseline (upper panel, *p* = 0.002). After nutritional intervention (NI), the gut microbiota of OB did not segregate from that of NW (lower panel, *p* = 0.122). (**b**) At baseline, the microbial communities of OB patients and NW controls present distinct signatures. Results are ranked by the Linear Discriminant Analysis value (LDA score): bacteria in red were more abundant in OB, while bacteria in green were more abundant in NW. (**c**) The NI reversed some of the microbial patterns identified at baseline.

**Figure 2 nutrients-12-02707-f002:**
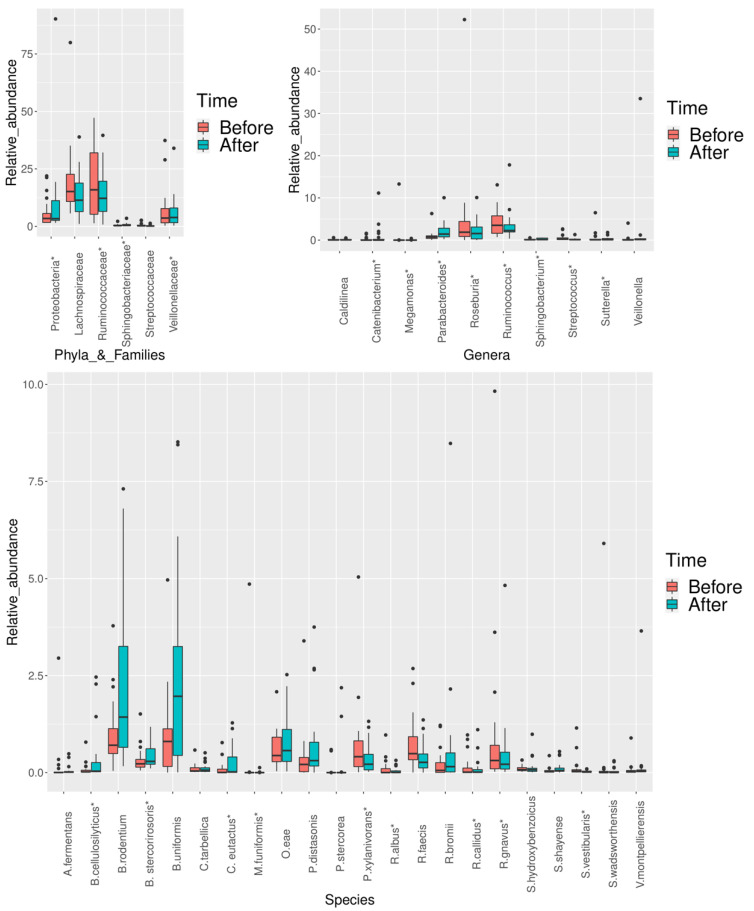
Statistically significant differences in bacterial relative abundance in obese and overweight patients after the nutritional intervention, at the phylum, family, genus, and species levels, respectively. The significance level was obtained by performing the Wilcoxon test for paired data. *p*-values were adjusted for False Discovery Rate (FDR) (FDR < 0.05). A change in twenty-four taxa was found when considering the FDR adjustment (as indicated by the asterisk). In the box plots, the boundary of the box closest to zero indicates the 25th percentile, a black line within the box marks the median, and the boundary of the box farthest from zero indicates the 75th percentile. Whiskers above and below the box indicate the 10th and 90th percentiles. Points above and below the whiskers indicate outliers outside the 10th and 90th percentiles. Each group is identified by colors, as indicated on the right side of the figure (before intervention = pink, after intervention = light blue). Every sample is represented by a black dot.

**Figure 3 nutrients-12-02707-f003:**
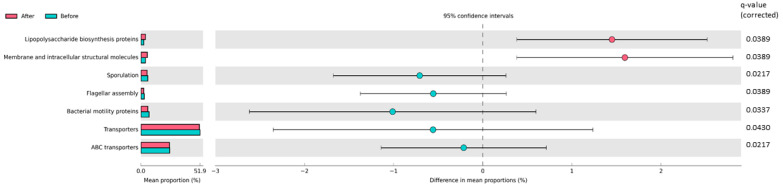
Statistically significant differences in predicted metabolic pathways in obese and overweight patients after the nutritional intervention (NI). The OTUs table generated by QIIME for the bacterial communities was analyzed by using the PICRUSt algorithm. Seven metabolic pathways were significantly different after NI. Pathways that were more abundant after NI are on the positive side (pink circle with 95% CI). Pathways that were less abundant after NI are on the negative side (light-blue circle with 95% CI). The q-values represent Benjamini–Hochberg’s FDR-corrected *p*-value (Wilcoxon test for paired data). Mean proportions are shown in stacks (before NI = light-blue; after NI = pink). The difference in mean proportions indicates the mean proportion after NI minus the mean proportion before NI.

**Table 1 nutrients-12-02707-t001:** Clinical characteristics of the recruited participants.

Clinical Parameter	OB	NW
Sex, (Females/Males)	20/3	40/6
Age, M (SD)	53 (9)	49 (11)
Overweight, N (%)	3 (13)	0
Class 1 obesity, N (%)	8 (35)	0
Class 2 obesity, N (%)	9 (39)	0
Class 3 obesity, N (%)	3 (13)	0
Hypertension, N (%)	7 (30)	0
Dyslipidemia, N (%)	6 (26)	0
Insulin Resistance, N (%)	3 (13)	0
Type II Diabetes, N (%)	2 (9)	0
Current Smoking status (Yes), N (%)	6 (26)	7 (15)
Former Smoking status (Yes), N (%)	5 (22)	0
Current Alcohol consumption (None), N (%)	9 (39)	10 (22)
Current Alcohol consumption (Rare), N (%)	8 (35)	24 (52)
Current Alcohol consumption (Moderate), N (%)	6 (26)	12 (26)

M—mean; N—number, SD—Standard deviation. For alcohol consumption: None—abstention from alcohol; Rare—occasional drinker; Moderate ≤ 300 mL per day. OB – obese; NW – normal weight.

**Table 2 nutrients-12-02707-t002:** Anthropometric measurements and nutritional intake in normal-weight controls (NW) at baseline and in overweight and obese patients (OB) before and after the nutritional intervention.

	NW	OB		
Clinical Parameter	T0	T0	T3	*p* *
Weight (Kg), M (SD)	54.9 (11.0)	89.5 (19.3)	82.8 (17.0)	0.015
Waist circumference (cm), M (SD)	74 (6)	108 (14)	102 (16)	0.040
Body mass index, M (SD)	21.6 (2.0)	35.2 (4.3)	33.6 (4.5)	0.001
Fat mass (Kg)	NA	37.8 (10.2)	32.7 (8.2)	0.0002
Muscle mass (Kg)	NA	47.2 (14.0)	47.6 (9.8)	0.493
Daily caloric intake (Kcal), M (SD)	1468 (160)	1779 (534)	1341 (298)	0.007
Carbohydrates intake (%), M (SD)	51 (3)	50 (6)	50 (8)	0.578
Lipids intake (%), M (SD)	27 (4)	33 (6)	29 (9)	0.196
Saturated lipids intake/Total lipids intake (%), M (SD)	28 (4)	39 (5)	35 (8)	0.139
Daily proteins intake (grams/day), M (SD)	62 (9)	73 (23)	64 (13)	0.384
Daily fibers intake (grams/day), M(SD)	20 (3)	14 (6)	17 (6)	0.234
MedDietScore **	33 (4)	29 (5)	32 (5)	0.665

* The difference after the nutritional intervention (NI) in OB was tested using a *t*-test for paired data. ** The MedDietScore was significantly higher in NW than in OB at T0 (*p* = 0.001) but not at T3 (*p* = 0.162). M—mean; NA—Not available; SD—Standard deviation.

**Table 3 nutrients-12-02707-t003:** Significant changes in the bacterial relative abundance after the nutritional intervention in overweight and obese patients.

Phylum	Family	*Genus*	*Species*	Median (IQR) at T0	Median (IQR) at T3	Prevalent Direction of Change (N)	*p*	q
Actinobacteria	Bifidobacteriaceae	*Bifidobacterium*	*B. bifidum*	0 (0.002)	0.006 (0.073)	↑ (15)	0.043	0.196
Bacteroidetes	Bacteroidaceae	*Bacteroides*	*B. cellulosilyticus*	0.019 (0.050)	0.036 (0.238)	↑ (20)	0.006	**0.039**
			*B. rodentium*	0.706 (0.643)	1.431 (2.603)	↑ (16)	0.012	0.059
			*B. stercorirosoris*	0.226 (0.199)	0.289 (0.400)	↑ (16)	0.016	0.091
			*B. uniformis*	0.803 (0.970)	1.967 (2.808)	↑ (17)	0.005	**0.036**
	Tannerellaceae	*Parabacteroides*		0.742 (0.689)	1.485 (1.646)	↑ (16)	0.016	0.093
			*P. distasonis*	0.211 (0.367)	0.309 (0.613)	↑ (17)	0.019	0.097
	Prevotellaceae	*Prevotella*	*P. stercorea*	0 (0)	0 (0.001)	≡ (14), ↑ (8)	3.09 × 10^−5^	**0.001**
	Sphingobacteriaceae			0.302 (0.389)	0.442 (0.454)	↑ (13)	1.62 × 10^−4^	**0.003**
	Sphingobacteriaceae	*Sphingobacterium*		0.088 (0.119)	0.115 (0.308)	↑ (12)	1.26 × 10^−3^	**0.011**
			*S. shayense*	0.039 (0.042)	0.055 (0.083)	↑ (16)	1.62 × 10^−4^	**0.003**
Chloroflexi	Caldilineaceae	*Caldilinea*		0.042 (0.097)	0.063 (0.096)	↑ (16)	0.045	0.196
			*C. tarbellica*	0.042 (0.097)	0.063 (0.096)	↑ (14)	0.045	0.196
Firmicutes	Acidaminococcaceae	*Acidaminococcus*	*A. fermentans*	0.003 (0.009)	0.018 (0.025)	↑ (14)	0.036	0.099
	Erysipelotrichaceae	*Catenibacterium*		0.002 (0.113)	0.005 (0.366)	↑ (19)	0.007	**0.049**
	Lachnospiraceae			15.329 (10.160)	11.358 (13.241)	↓ (14)	0.042	0.194
	Lachnospiraceae	*Coprococcus*	*C. eutactus*	0.004 (0.089)	0.018 (0.399)	↑ (20)	0.001	**0.005**
		*Pseudobutyrivibrio*	*P. xylanivorans*	0.410 (0.665)	0.216 (0.402)	↓ (17)	5.23 × 10^−5^	**0.001**
		*Roseburia*		1.904 (2.995)	1.379 (2.298)	↓ (19)	0.004	**0.026**
		*Roseburia*	*R. faecis*	0.489 (0.594)	0.264 (0.360)	↓ (16)	2.70 × 10^−5^	**0.001**
Firmicutes	Selenomonadaceae	*Megamonas*		0 (0.002)	0 (0)	≡ (14), ↓ (9)	0.007	**0.046**
		*Megamonas*	*M. funiformis*	0 (0.002)	0 (0)	≡ (13), ↓ (10)	0.005	**0.038**
	Ruminococcaceae			15.395 (23.822)	13.491 (14.593)	↓ (12)	2.62 × 10^−4^	**0.003**
	Ruminococcaceae	*Oscillospira*	*O. eae*	0.440 (0.632)	0.568 (0.825)	↑ (14)	0.048	0.196
		*Ruminococcus*		3.561 (4.819)	2.284 (2.619)	↓ (12)	2.70 × 10^−5^	**0.001**
			*R. albus*	0.001 (0.102)	0.005 (0.045)	↓ (13)	0.003	**0.012**
			*R. bromii*	0.056 (0.252)	0.154 (0.491)	↑ (12)	0.039	0.186
			*R. callidus*	0.011 (0.116)	0.013 (0.074)	↓ (12)	0.001	**0.008**
			*R. gnavus*	0.312 (0.607)	0.215 (0.433)	↓ (15)	0.042	0.196
	unclassified Tissierellia	*Sedimentibacter*	*S. hydroxybenzoicus*	0.075 (0.078)	0.073 (0.082)	↓ (13)	3.33 × 10^−4^	**0.004**
	Streptococcaceae			0.218 (0.329)	0.114 (0.136)	↓ (17)	0.009	0.051
		*Streptococcus*		0.215 (0.332)	0.114 (0.134)	↓ (16)	0.015	0.073
		*Streptococcus*	*S. vestibularis*	0.038 (0.060)	0.014 (0.029)	↓ (20)	3.09 × 10^−5^	**0.001**
	Veillonellaceae			3.338 (6.828)	3.492 (6.288)	↓ (12)	2.95 × 10^−4^	**0.004**
	Veillonellaceae	*Veillonella*		0.059 (0.123)	0.150 (0.185)	↑ (17)	3.73 × 10^−4^	**0.004**
			*V. montpellierensis*	0.024 (0.047)	0.038 (0.048)	↑ (15)	1.27 × 10^−4^	**0.001**
Proteobacteria				3.492 (3.670)	3.502 (5.981)	↓ (12)	2.70 × 10^−5^	**0.001**
	Sutterellaceae	*Sutterella*		0.108 (0.187)	0.204 (0.420)	↓ (12)	0.001	**0.012**
			*S. wadsworthensis*	0.002 (0.033)	0.003 (0.029)	↑ (10)	0.018	0.093

Median = median values of the bacterial relative abundance before (T0) and after (T3) nutritional intervention (NI) (bacterial relative abundance is expressed as percentage from 0 to 100); IQR = interquartile range calculated as the difference between upper and lower quartiles; N = number of patients that reported the most prevalent direction of change in the bacterial relative abundance (↓ = reduced after NI, ↑ = increased after NI, ≡ = no different after NI). Results were obtained by the Wilcoxon test for paired data performed on R software (v. 3.5.2). q = *p* adjusted for Benjamini and Hochberg’s false discovery rate (FDR) correction test for multiple comparisons (FDR < 0.05). ↓ = significantly reduced after the NI, ↑ = significantly increased after the NI. Bold values denote statistical significance (q ≤ 0.05).

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
