# Peer review of "Impact of a Moderately Hypocaloric Mediterranean Diet on the Gut Microbiota Composition of Italian Obese Patients"

_nutrients, 2020, doi:10.3390/nu12092707_

Round 1

Reviewer 1 Report

Pisanu and colleagues evaluate the effects of 3 months’ consumption of a low-calorie Mediterranean diet on the gut microbiota profile of 23 overweight/obese adults (BMI>25). A lean control group is compared at baseline but does not undergo the diet intervention. The diet intervention improves anthropometric parameters and alters composition of the gut microbiota.

  1. Because the OB group contains a wide range of BMIs, it would be prudent to include BMI as a covariate when assessing changes in anthropometric parameters (Table 2). Thus, these would need to become within-subjects ANOVAs. The same applies to analyses of gut microbiota composition. For example, the F:B ratio (line 208) is highly variable, and I wonder whether individual differences in weight may contribute to this.
  2. Please clarify whether meals were provided, or whether participants were instructed to follow the prescribed plan. It would be helpful to provide more detail about the foods eaten by participants, because it is surprising that alpha diversity did not differ between OB/control groups at baseline, and did not change in the OB group over time.
  3. A significant limitation is that the lean control group did not undergo the diet intervention. This leads to difficulty in interpretation. For example, Figure 1a shows 2 NMDS plots: one comparing control and OB groups at baseline, and another comparing the same data from the control group with the OB group after the diet. These are used to show that the intervention improved the biome of OB patients. However the authors have not directly the OB group before and after the intervention. This should be reported.
  4. Is it surprising that the Med diet score did not change significantly across the intervention? What might account for this?
  5. Line 167-169: more information is needed about the food diaries – please report summary information for both groups at baseline, and for the OB group at follow-up.
  6. Several relevant papers on the gut microbiota and Mediterranean diet have not been discussed, including (but not limited to):

Adherence to the Mediterranean diet is associated with the gut microbiota pattern and gastrointestinal characteristics in an adult population (Mitsou et al. 2017 Brit J Nutr, 117(12), 1645-1655)

Mediterranean diet and faecal microbiota: a transversal study (Gutierrez-Diaz et al. 2016 Food & Function, 7, 2347-2356)

Mediterranean Diet and Health: Food Effects on Gut Microbiota and Disease Control (Del Chierico et al. 2014 Int J Mol Sci, 15(7), 11678-11699)

High-level adherence to a Mediterranean diet beneficially impacts the gut microbiota and associated metabolome (De Filippis et al. 2015, Gut, 65(11), 1812-1821)

Minor comments

Line 47: “the development and homeostasis in adult life” is unclear. Development and homeostasis of what?

Line 182: a word is missing after ‘nutritional’ at the end of the line.

Line 192: ‘noteworthy’ is better written ‘notably’

Line 286: typo – ‘overweigth’

Reviewer 2 Report

The ms describes changes in faecal microbiota of obese/overweight patients due to moderately hypocaloric Mediterranean diet.

The authors identified (a) changes in the relative abundance of bacterial taxa during the diet and (b) differences in microbiota structure between the obese/overweight patients and lean controls.

The authors may wish to consider the following points:

Major

  1. There are many grammatical and syntactical errors. The ms would benefit from additional proof-reading and editing for linguistic clarity.
  2. The data from the control group should be added into Tables 1 (except for some parameters) & 2.
  3. There is a strong gender bias (20 females vs 3 males). Were the samples from men clustered in Figure 1a? Did they represent obvious “outliers”?

The figure 1bc and Table 3 could be generated (but not necessarily added to the ms) taking into consideration only female participants. Depending on the results of these analyses, the authors may briefly state in the Results or Discussion section that they obtained identical or very similar results when analysing only samples from females, or, that additional biomarkers were discovered using this subset.

  1. Line 61 & 62. Lactobacillus and Bifidobacterium are not butyrate producers but they interact with bacteria producing butyrate.
  2. Line 167. It would be appropriate to provide per-patient and per-control metadata (including nutritional) in a supplementary table.

Minor

Line 53. Define (full name) AMPK

Line 84. Lowercase for “Intestinal Bowel Disease”

Line 92-93. End the sentence with “stadiometer”. Details about shoes and head are not relevant.

General. Capitalise words in headings, as per journal style

General. Use uniformly spaces around “>”, “<” and “=”, as per journal style.

General. Use uniformly “V.” or “v.” (but not both)  for version of software.

General. Instead of “p-value” only “p” can be used.

Line 115. Please state briefly what LARN refers to.

Line 121-123. These details had already been provided in the reference quoted [16].

Line 126. The primers used target the V3–V4 region rather than V3 and V4 regions.

Line 129. MiSeq

Line 132-133. It is slightly unclear what “using a two-step open-reference operational taxonomic unit (OTU)” means. “two-step picking”, “two-step process”, or other?

Line 139. “taxonomic profiles” rather than “taxonomic levels”

Line 150. Indicate the version of PICRUSt used.

Line 167. “anamnesis”.

Table 1. How was the alcohol consumption level categorised (Winfood?)? This may be Indicated in the methods section (Line 102-109).

Line 174-175. Use a point as decimal separator.

Line 194. “beta diversity” instead of “Beta diversity”

Line 198. “upper panel” instead of “upper figure”

Line 199-200. “Obese and overweight patients present distinct microbial signatures at baseline” is slightly confusing.  Did you mean “at baseline, microbial communities of OB patients and NW controls present distinct signatures”?

Line 201. “expressed” is not appropriate, use “abundant”

Line 205. The sentence can be modified, e.g. “… had a Firmicutes/Bacteroidetes ratio more than twice that of controls”

Line 207. Delete “after NI”

Line 225-226. End the sentence after “strength”, remove the rest which is slightly confusing.

Figure 2. Present phylum, family and genus data as three separate horizontally aligned panels (first raw) and species-level data in a second row. Consequently, the text in Line 239 will read “… phylum, family, genus and species…”

Line 242-243. This sentence should be re-written for clarity. The length of the whisker has not been explained.

Line 246. Table heading should include the mention of OB patients

Table 3. Please avoid using the epithet alone for species names (e.g. write “B. bifidum” instead of “bifidum”). Try also to use smaller font size to avoid table continuation. Explain what values in bold represent (q < 0.05).

Line 250. “q” (lowercase) instead of “Q”.

Line 253. “observed” rather than “obtained”.

Line 313-315. What is the putative role of Bacteroidetes members in these mechanisms? Please clarify of modify the sentence.

Line 318-319. “… reversion of some of the signatures previously identified …” What does “previously identified” refer to? Please cite the relevant article. Or, perhaps, you refer to the present study?

Line 326. “gut microbial alterations” is not clear. Did the authors mean “gut microbial changes during NI”?

Line 346. “provided by this family” can be replaced, for example, by “provided by several members of this family”.

Line 352-353. Please cite the paper related to SCFA production by these bacteria.

Line 360. “Oscillospiraceae”

Line 373. “with”

Line 382. Remove space after “Alistipes”

Line 388. Increased cell motility has also been observed in high-fat diet-fed mice (Everard et al., 2014, The ISME Journal 8, 2116-2130).

Line 388-389. Replace “In contrast with the present work” by “However”.

Line 391-393. LPS biosynthesis pathway was not associated with obesity in Scientific Reports 8: 9749 (2018)

Line 392. Please check whether the reference 49 (leukaemia patients) is appropriate.

Round 2

Reviewer 2 Report

The authors have carefully revised and improved the MS.

There are a couple of points to be considered before paper acceptance:

Methods (and supplementary tables): No mention of ANOVA global and/or post hoc tests used.

Line 57: Remove “To date” or “to this day” (or both because you talk about an open challenge).

Line 185 (Table 2): NW column: TO should read T0

Line 192: “Similarly, no significant change in this index (3.39 ± 0.66, p = 0.761) was found when comparing T0 and at T3 samples of OB patients”.

Line 193: “Differences in the Shannon index across different BMI categories were not statistically significant, both at T0 and T3…”

Line 196: “dividing the total study population by”

Line 201. “although statistical significance”

Line 223. “the taxa that were identified as differentially abundant using the KW”

Line 236: “differences in the Firmicutes/Bacteroidetes ratio and in the relative abundance of same taxa within these phyla were no longer significant;”

Line 236-240: This phrase starting with “in particular,” may be deleted

Line 241-242: “as indicated by the paired analysis” may be deleted

Line 244 “OB (at T0 or T3) and NW”

Line 244: “the matched T0–T3 OB analysis”

Line 245: “were performed” instead of “was repeated”

Table 3: Two p values = 0.000. Please indicate the exact values.

Line 265: Replace “At the phylum level,” by “Following NI”

Line 335: What does FXR stand for?

Line 359: “bacterial relative abundance”

Line 382: Remove “Both”

Line 396: “is among the most involved” is a bit unclear

Line 402: remove “published last year”

Line 442: “decrease in gene functions related to membrane transport”

Line 462: Do the authors mean “microbiota variability among the included patients” ?

Tables S2 and S4: Please check whether these p-values are correct (identical values in the two tables).

Tables 3 and S5. Acidaminococcus fermentans significantly increased when 23 subjects were analysed (Table S3), but significantly decreased when considering 20 females (Table S5). The authors probably defined increase/decrease by the positive/negative sign of MD (mean difference of relative abundance”. If this was the case, it might be better to consider the prevalent direction of change (number of subjects in which this direction i.e. increase or decrease) occurred and report this value in parenthesis next to the arrow (in the column between “MD” and “p”). Instead of MD, median and IQR values may be presented (because a non-parametric test was used). Finally, MD also stands for “Mediterannean diet” in the ms, better not to use the same abbreviation.

Table S5. “female instead of “Female”
